# MANGACRAFTER:
# TRAINING-FREE CONSISTENT MANGA GENERATION VIA PHASED DIFFUSION

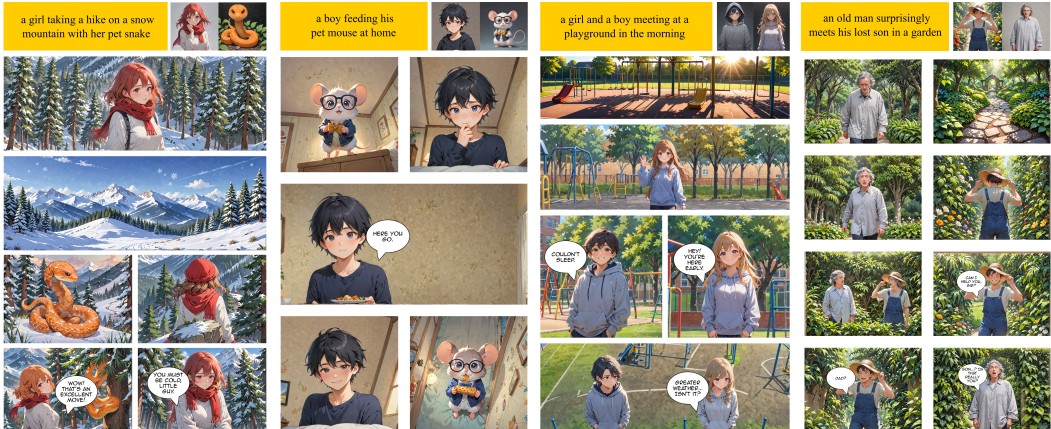

Figure 1: **MangaCrafter** achieves a strong balance between character consistency and prompt alignment, producing diverse imagery that follows the artistic and evolving storylines of the input prompt. Notably, consistent characters are not limited to humans but extend to a wide range of entities. Reading order follows Japanese manga convention: top to bottom, right to left.

## ABSTRACT

The generation of consistent characters across an entire manga page is important yet challenging, as characters must remain coherent under diverse poses, actions, and layouts. Unlike conventional face or human consistency methods that focus on isolated portraits, this broader narrative setting cannot be directly addressed by per-subject fine-tuning or narrowly scoped identity-preservation techniques. We introduce **MangaCrafter**, a 3-phase training-free framework that achieves layout-aware, multi-character manga generation by altering the denoising processes of latent diffusion. Our key insight is that character consistency can be secured not through persistent identity injection but through a phased control of the diffusion trajectory that front-loads identity anchoring while gradually relaxing constraints to enable expressive, prompt-driven detail. In **Phase 1**, *Structural Resonance Injection (SRI)* augments the UNet's attention with cached reference features to robustly establish structural similarity in the high-noise regime. The centerpiece of our contribution lies in **Phase 2**, where the *Predictive Drift Controller (PDC)*, a proportional-integral-derivative feedback system, dynamically measures feature drift between the evolving latent and the reference to modulate the denoising process, ensuring robust identity preservation while suppressing "pasted-on" and "blurry" artifacts. Finally, in **Phase 3**, we strategically zero out reference injections, transferring identity control to the early imprints while allowing the model to synthesize fine, prompt-driven details without over-similarity. Together with a lightweight preprocessing workflow that resolves multi-character fusion, MangaCrafter delivers training-free, consistent yet flexible manga synthesis and suggests a general paradigm for controlled narrative generation across diffusion-based media. Extensive experiments on the challenging ConsiStory+ benchmark show that our framework achieves state-of-the-art identity preservation while maintaining high prompt alignment. Ablations confirm the effectiveness of our phased design in balancing consistency, diversity, and aesthetic quality.

## 1 INTRODUCTION

The advent of large-scale text-to-image diffusion models has transformed digital content creation, enabling the synthesis of photorealistic and stylistically diverse imagery from natural language descriptions. This capability has sparked growing interest in automated visual narrative generation, which requires not only high-quality image synthesis but also coherence and consistency across image sequences. Among various narrative forms, manga (Figure 1), with its distinctive artistic style, complex panel layouts, and emphasis on expressive character arcs, poses a particularly difficult challenge that remains largely unmet by current generative frameworks.

Existing text-to-image models excel at generating isolated images but struggle with the sequential and relational demands of manga creation. A central challenge is maintaining character consistency, as state-of-the-art models often fail to preserve identity, including facial features, attire, and overall appearance, across panels with varied poses, expressions, and actions. Conventional methods for enforcing identity typically face a consistency-alignment trade-off: they either preserve the character too rigidly, restricting alignment to prompt-driven actions or emotions, or they prioritize the prompt at the expense of the character's core, consistent identity.

Prior work can be categorized into training-based and training-free approaches. Training-based methods, such as DreamBooth (Ruiz et al., 2023) and textual inversion (Gal et al., 2022), fine-tune models to learn a new concept for a specific character. These methods are effective at identity preservation but are computationally expensive, require per-subject optimization, and risk overfitting to limited reference images, limiting generalization to novel contexts. Training-free methods offer greater efficiency but still encounter significant trade-offs. For example, StoryDiffusion (Zhou et al., 2024a) may fail to maintain strong identity adherence especially when the generated image's aspect ratio deviates largely from that of the reference image, while One-Prompt-One-Story (Liu et al., 2025) can produce over-similar results, especially if the reference image primarily consists of large, spatial features, rigidly copying the reference pose.

To address these limitations, we propose **MangaCrafter**, a 3-phase, training-free framework for layout-aware, multi-character manga generation. The core idea is to maintain character consistency not through persistent identity injection but through phased control of the denoising processes, front-loading identity anchoring while gradually relaxing constraints to allow prompt-driven expression.

**Phase 1: Structural Resonance Injection (SRI).** In the early, high-noise timesteps ($t > T_{phase1}$), SRI manipulates the UNet's self-attention mechanism. For each attention block, the query derived from the current noisy latent attends to key and value matrices formed from the concatenation of the current latent features and precomputed reference character features, imprinting fundamental structural and visual attributes.

**Phase 2: Predictive Drift Control (PDC).** Once the core structure is established ($T_{phase2} < t \leq T_{phase1}$), the PDC computes feature-space drift between the current latent and a noisy reference latent using an $L_1$ loss. This drift is fed into a proportional-integral-derivative (PID) controller, producing a modulation factor that dynamically adjusts latent blending. The result is continuous correction of identity without the rigidity of direct feature injection, while avoiding artifacts such as blurriness or a pasted-on appearance.

**Phase 3: Refinement and Zero-Out.** In the final low-noise timesteps ($t \leq T_{phase2}$), reference injections are zeroed out, transferring identity control to the early phases and allowing the model to synthesize fine, prompt-driven details without over-similarity. This phase ensures that characters remain consistent while enabling expressive variation crucial for narrative progression.

Our contributions are summarized as follows:

- We present a novel, end-to-end training-free framework for customized manga generation that jointly addresses layout complexity and robust character consistency.
- We introduce a multi-phase generation process with Structural Resonance Injection (SRI) and Predictive Drift Control (PDC), the first application of PID control theory to the diffusion feature space for identity preservation, establishing a new paradigm for controllable generation.
- Extensive experiments on the ConsiStory+ benchmark demonstrate state-of-the-art performance, showing that our framework outperforms existing training-free methods in charac-

ter consistency, prompt alignment, and image quality, while ablations confirm the effectiveness of the phased design in balancing identity, diversity, and aesthetics.

## 2 RELATED WORK

**Consistent Generation.** Achieving consistent character identity and spatial coherence across multiple generated images without costly fine-tuning is a key challenge in generative modeling. A common strategy is manipulating attention mechanisms in pre-trained diffusion models. ConsiStory (Tewel et al., 2024) introduces a Subject-Driven Self-Attention (SDSA) block that lets images attend to subject-specific patches in other frames via a mask. While effective, masking limits diversity and often yields a "pasted-on" look. StoryDiffusion (Zhou et al., 2024a) is similarly constrained, maintaining consistency for only one character per panel and failing with multi-character interactions central to manga. It is also brittle: deviations in panel aspect ratio cause major quality degradation and identity loss. MasaCtrl (Cao et al., 2023) also uses mutual self-attention without training, but targets editing rather than narrative variation. For manga, where diverse poses, angles, and expressions are essential, such restrictions are severe. Identity-preserving methods such as ID-Booth (Tomašević et al., 2025) and ID$^3$ (Li et al., 2024) focus on faces, while CoDi (Gao et al., 2025) extends consistency to varied poses. Other approaches include IP-Adapter (Ye et al., 2023), which injects image features via cross-attention; The Chosen One (Avrahami et al., 2023), which clusters large image sets for identity distillation; and One-Prompt-One-Story (Liu et al., 2025), which concatenates prompts to exploit text encoder self-attention. StoryMaker (Zhou et al., 2024b) further enforces consistency across characters, clothing, and environments in story-driven generation.

Other recent training-free approaches, such as CharacterFactory (Wang et al., 2024), FastComposer (Xiao et al., 2023), OPT2I (Mañas et al., 2024), and CharaConsist (Wang et al., 2025), use latent-space control, prompt optimization, or fine-grained feature alignment to improve identity consistency, but often trade off diversity or require iterative refinement. Unlike prior work, MangaCrafter implements phased control along the diffusion trajectory, embedding a physics-inspired mechanism directly into the generation process. This approach provides adaptive, phase-wise regulation of both identity and spatial layout, without masking, iterative refinement, or rigid prompt concatenation. It enables higher identity consistency while maintaining the expressive flexibility critical for manga storytelling.

**Manga and Layout Generation.** Manga generation poses unique challenges due to complex panel layouts and distinct visual grammar. Early methods focused on style transfer rather than de novo generation (Zhou et al., 2024a). Recent approaches such as DiffSensei (Wu et al., 2024) integrate a Multimodal Large Language Model (MLLM) as a text-compatible identity adapter, masked cross-attention, and dialog layout embedding, enabling fine-grained control over character poses, expressions, and interactions within panels. MangaDiffusion (Chen et al., 2024) uses transformer-based intra- and inter-panel blocks to manage both panel content coherence and flexible page layouts, and introduces the Manga109 dataset for training and evaluating layout controllable multi-panel manga generation. Studies like "How Panel Layouts Define Manga" (Feng et al., 2024) investigate the structural importance of panel layout itself, showing that panels' spacing, alignment, and ordering encode stylistic and narrative cues that are characteristic of manga works.

Our layout module uses the training-free LayoutPrompter (Lin et al., 2023), which leverages the MangaZero dataset in retrieval-and-composition manner. This enables diverse and coherent page layouts without requiring learned generative models. Unlike DiffSensei and MangaDiffusion, which are trained generative models for layout control, our method avoids large-scale training for layout generation, reducing computational costs while maintaining layout diversity and coherence aligned with narrative structure.

**Control Theory in Generative Models.** Integrating classical control principles into deep generative models is an emerging area. Proportional-Integral-Derivative (PID) controllers (Åström & Hägglund, 1995) are foundational in control systems, known for minimizing the error between a measured variable and a desired setpoint while providing robustness and stability. Prior works such as RCDM (Xu et al., 2024) and optimal control perspectives on diffusion-based models (Berner et al., 2022) explore using control-theoretic ideas to guide stochastic generation. To our knowledge, MangaCrafter's Predictive Drift Controller (PDC) is the first to apply a PID-like loop directly in the feature space of a diffusion model for identity preservation. By treating character identity as the setpoint and feature-space divergence as the error, the PDC enables adaptive, fine-grained control that

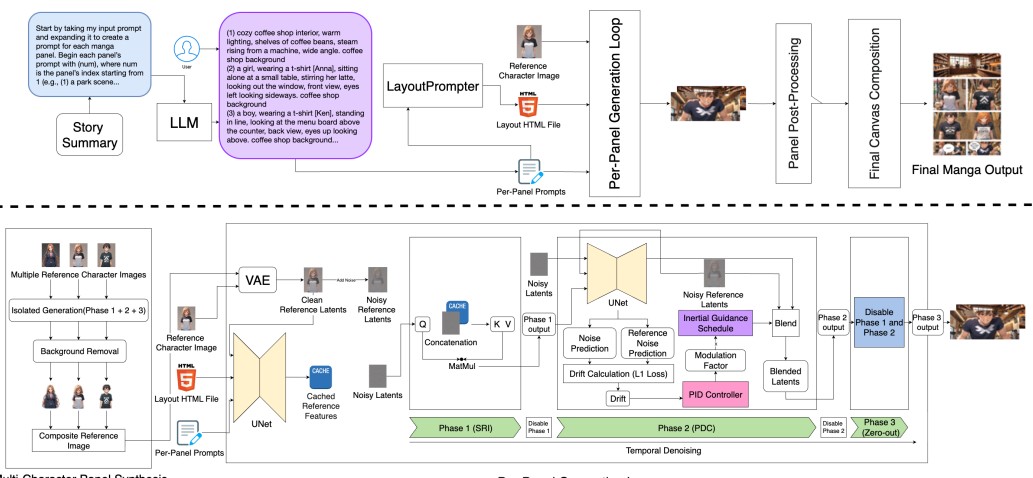

Figure 2: **The pipeline of MangaCrafter.** Top shows the overall pipeline; bottom is a detailed illustration of the per-panel generation loop. In the top figure, the process in the box labeled "Panel Post-Processing" adds dialog bubbles onto the generated manga panels. In the bottom figure, the blue box in Phase 3 represents the "zero-out" operation. Please zoom in for details.

corrects past deviations and anticipates future drift, offering a more robust and flexible alternative to static attention mechanisms.

# 3 METHOD

In this section, we present the pipeline of our training-free manga generation framework, which combines layout generation with a multi-phase process to ensure prompt-adherent character consistency. We first obtain input prompts for each manga panel either from raw user inputs or LLM-generated results. We then employ LayoutPrompter (Lin et al., 2023), a training-free retrieval and composition module, to generate page layouts from a story description. Leveraging the MangaZero dataset (Wu et al., 2024), LayoutPrompter analyzes narrative requirements such as character count and scene transitions, retrieves analogous layouts, and composes new page structures. The output provides precise panel coordinates along with bounding boxes for characters and dialogue, serving as a structural guide for subsequent image generation. This training-free approach enables diverse, conventional manga layouts without costly fine-tuning or architectural modification.

The core of our contribution to identity preservation is a multi-phase generation process that dynamically manages the influence of a reference character image throughout the denoising process (Figure 2). This method ensures robust identity replication in the early, formative phases of generation while allowing for prompt-driven flexibility and refinement in the later phases.

## 3.1 PHASE 1: STRUCTURAL RESONANCE INJECTION (SRI)

In the initial high-noise timesteps, where $t > T_{phase1}$, the primary objective is to imprint the fundamental structural and visual characteristics of the reference character onto the canvas. To achieve this, we introduce a custom attention mechanism. First, the reference character image is processed through the UNet to extract and cache its intermediate self-attention features, $h_{ref}$.

During the generation of the target image, for each self-attention layer within the UNet, we augment the Key ($K$) and Value ($V$) projections. While the Query ($Q$) is derived solely from the current noisy latent's hidden states, $h_{current}$, the Key and Value matrices are derived from the concatenation of $h_{current}$ and the cached reference features $h_{ref}$. This can be formulated as:

$$Q = W_Q \cdot h_{current}$$

$$K = W_K \cdot \text{concat}(h_{current}, h_{ref})$$

$$V = W_V \cdot \text{concat}(h_{current}, h_{ref})$$

This forces the generation to "resonate" with the reference character's features, ensuring core attributes like facial structure, hair, and attire are established early. This approach helps navigate the consistency-alignment trade-off. By separating the query from the keys and values, SRI disentangles the character's static identity from their dynamic, prompt-driven state. The query, driven by the target prompt (e.g., "a girl walking"), dictates *what* to generate, while the augmented key-value space provides the reference character's visual vocabulary for *how* it should be rendered. This enables selective feature retrieval, preserving identity while accurately executing the prompt, avoiding "concept bleed" where the reference pose overrides the target action.

## 3.2 PHASE 2: PREDICTIVE DRIFT CONTROL (PDC)

After the initial structure is set (for timesteps $T_{phase2} < t \leq T_{phase1}$), the process transitions from the aggressive attention manipulation of Phase 1 to a more nuanced guidance mechanism. This phase introduces the Predictive Drift Controller (PDC), a novel application of Proportional-Integral-Derivative (PID) control theory to the diffusion process.

The PDC operates as a closed-loop feedback system. At each step $t$, it first calculates the "drift," $d(t)$, which we define as the $L_1$ loss between the UNet's noise prediction for the target prompt, $\epsilon_\theta(z_t, t, c_{target})$, and a parallel prediction for the reference character, $\epsilon_\theta(z_t^{ref}, t, c_{ref})$. Here, $z_t^{ref}$ is the noisy version of the reference latent at the same timestep. The drift $d(t)$ quantifies the feature-space error between the current generation and the desired character identity.

This error is then fed into the PID controller, which calculates a modulation factor $M_{PDC}(t)$ to correct the generation trajectory:

$$P(t) = K_p \cdot d(t)$$
$$I(t) = I(t-1) + K_i \cdot d(t)$$
$$D(t) = K_d \cdot (d(t) - d(t-1))$$
$$M_{PDC}(t) = 1.0 + P(t) + I(t) + D(t)$$

where $K_p$, $K_i$, and $K_d$ are the proportional, integral, and derivative gains, respectively. This modulation factor does not act in isolation; it scales a schedule that we term the Inertial Guidance Schedule, $\mathcal{G}(t)$. It provides the foundational, non-reactive "momentum" for the guidance strength, and defines a pre-determined trajectory of influence that resists deviation, analogous to an object's inertia for maintaining its state of motion. The PDC then acts as the active, corrective force that adjusts this inertial path. For this schedule, we utilized a quadratic decay schedule:

$$\mathcal{G}(t) = \left(1.0 - \left(\frac{T_{start} - t}{T_{start} - T_{end}}\right)^2\right) \cdot \mathcal{G}_{max}$$

where $T_{start}$ and $T_{end}$ define the timesteps for Phase 2, and $\mathcal{G}_{max}$ is the maximum strength. This non-linear function ensures that the base guidance strength remains high and stable during the early phases of Phase 2, providing a robust foundation for the PDC to operate upon. As the denoising process approaches its final phases, the schedule's value drops off more rapidly, gracefully receding the influence of the reference latent.

Crucially, the PDC reduces visual defects that arise from an overly prolonged Structural Resonance Injection phase. If SRI persists too long, it not only creates a "pasted-on look" but also a distinct "blurriness," where the image appears unnaturally smoothed and aesthetically unpleasing. The PDC counters this with a regularising effect, removing the artifact and yielding high-quality images with strong reference similarity. This mechanism emerges from two complementary control signals: the macro-level trajectory of the Inertial Guidance Schedule and the micro-level, real-time corrections from the PDC. The blended latent $z_t^{blend}$ is then computed through this factor, subtly guiding generation back toward the reference identity at each step.

## 3.3 PHASE 3: STRATEGIC REFINEMENT AND LIBERATED SYNTHESIS

The final denoising phase, for timesteps $t \leq T_{phase2}$, represents a strategic transition from foundational control to creative refinement. A key innovation of our framework is the "zero-out" of the previous phases. This is not simply a shutdown of the identity preservation mechanisms, but a strategic and essential transfer of control. This strategic withdrawal is a direct testament to the efficacy

| Story Summary | Reference Prompts | 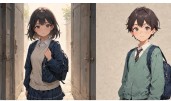 |
|---|---|---|
| A girl and a boy saying goodbye after school | [girl] a girl wearing a schoolbag. [boy] a boy wearing a schoolbag. | |

(1) a girl wearing a schoolbag, smiling and talking, "Me too, that test was crazy though!", side view, eyes right [girl], a boy wearing a schoolbag, laughing, side view, eyes left [boy], a tree-lined street during sunset.
(2) a boy wearing a schoolbag, smiling and talking, "Well, this is my way home.", [boy] a quiet intersection in a residential area.
(3) a girl wearing a schoolbag, a little sad, talking, "Oh, okay. See you tomorrow then.", front view, eyes down [girl], a quiet intersection in a residential area.
(4) a boy wearing a schoolbag, turning back, waving his hands while turning back, "Yeah! See you!" [boy], a quiet intersection in a residential area.
(5) The setting sun casting long shadows down the empty street, wide angle.
(6) The setting sun casting long shadows down the empty street, wide angle, "See you, Alex..." a quiet intersection in a residential area.

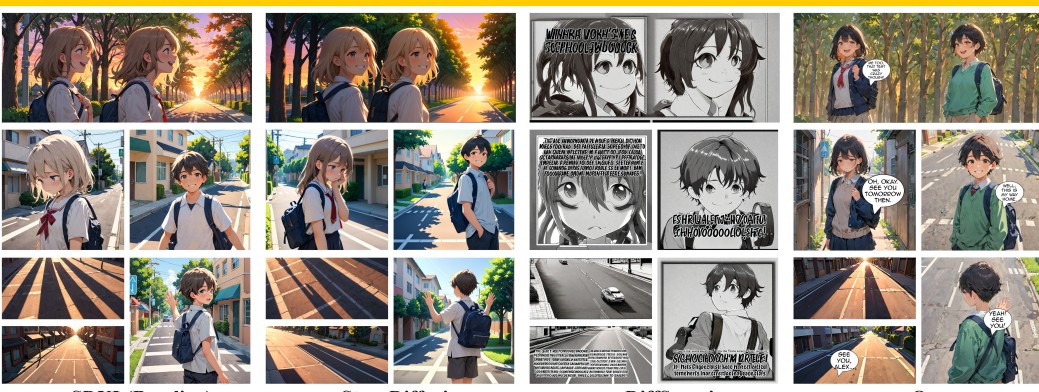

| SDXL(Baseline) | StoryDiffusion | DiffSensei | Ours |
|---|---|---|---|

Figure 3: **Qualitative Comparison.** Existing methods either enforce multiple characters to share the same appearance or fail to respect non-human entities (e.g. clothes), leading to identity collapse or missing elements. In contrast, our multi-phase framework maintains distinct character identities, captures diverse poses and expressions, and preserves compositional and aesthetic coherence across all elements in the scene.

of the preceding phases, where the structural imprinting from SRI and the continuous correction from the PDC are so robust at anchoring the character's core identity in the high-noise, high-impact early timesteps that persistent and heavy-handed intervention becomes not only unnecessary but counterproductive.

By ceasing direct attention manipulation, we free the model from the rigid constraints of reference features, allowing it to synthesize high-frequency, prompt-specific details such as subtle facial expressions, clothing textures under varied lighting, and nuanced emotional states. These details, absent in static references, are essential for dynamic storytelling. This phase mitigates the "over-similarity" problem common in consistency-focused methods, ensuring the character integrates naturally into the scene rather than appearing superimposed. This front-loaded efficiency, where identity is secured early in denoising, enables effective and controllable generation. Moreover, the SRI/PDC architecture shows promise for domains like text-to-video synthesis or complex scene composition, where establishing a core subject or style early while permitting temporal or spatial variation later maximizes both consistency and flexibility.

### 3.4 MULTI-CHARACTER PANEL SYNTHESIS

A common failure in multi-subject generation is "multi-character fusion," where distinct identities blend together. We address this, a limitation in prior works like DiffSensei (Wu et al., 2024), with a three-phase workflow for robust multi-character handling. First, each character specified in the prompt is generated individually at high resolution using the full generation pipeline, ensuring robust identity preservation. Second, backgrounds are removed from these individual character images, and the resulting foregrounds are composited onto a transparent canvas, arranged according to the layout's spatial coordinates. This produces a spatially coherent multi-character reference. Finally, the composite canvas guides a final generation pass with the full panel prompt and stronger

identity-preservation hyperparameters. This step integrates all characters naturally into the scene while preserving their unique identities, handling interactions, overlapping poses, and relative scale consistently.

# 4 EXPERIMENTS

## 4.1 EXPERIMENTAL SETUPS

**Comparison with SOTA methods.** We compare our method against a suite of state-of-the-art training-free consistent generation approaches, including One-Prompt-One-Story, StoryDiffusion, DiffSensei, and Consistory. We also include the base SDXL model as a performance baseline. To ensure a comprehensive and challenging evaluation, we utilize 1000 prompts from the extensive ConsiStory+ benchmark, as introduced by Liu et al. (2025).

**Evaluation Metrics.** We compute the CLIP-T (Hessel et al., 2022) for evaluating prompt alignment and utilizing two metrics for the evaluation of identity consistency: CLIP-I (Hessel et al., 2022) and DreamSim (Fu et al., 2023), which has been shown to correlate strongly with human perceptual judgment. To provide a quantitative measure of visual quality, we also employed the Fréchet Inception Distance (FID) (Heusel et al., 2018) to assess the aesthetic quality of the generated images.

## 4.2 EXPERIMENTAL RESULTS

**Qualitative Comparison.** Qualitative comparisons in Figure 3 and Figure 6 illustrate the practical advantages of our framework over existing methods. Existing approaches often enforce multiple characters to share the same appearance or fail to respect non-human entities, such as clothing, props, or non-human characters, which can result in identity collapse or missing elements. In contrast, our multi-phase framework maintains distinct character identities, captures diverse poses and expressions, and preserves compositional and aesthetic coherence across all elements in the scene.

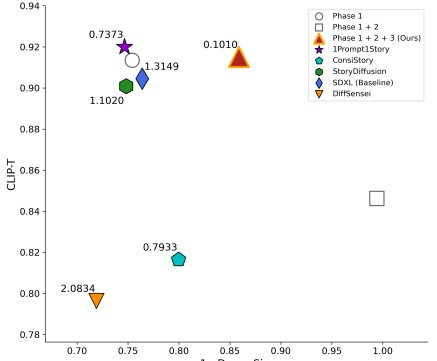

Figure 4: Trade-off between identity preservation and prompt alignment. Points near the upper-right indicate better balance; our method achieves the best overall balance with the lowest FID, outperforming other approaches.

Our multi-phase process achieves substantially better character consistency and yields a richer diversity of poses and expressions, which is vital for narrative progression, compared to other approaches. Furthermore, our framework demonstrably produces images with superior aesthetic appeal, balanced panel composition, and coherent spatial relationships between characters and objects. This ensures that each generated scene not only faithfully represents individual character traits but also maintains the integrity of interactions and narrative context, highlighting the expressive flexibility and robustness of our approach over prior work.

**Quantitative Comparison.** Figure 4 shows that our method achieves state-of-the-art performance in striking a balance between identity preservation, textual alignment, as well as visual quality, outperforming all other approaches. The more towards the upper right corner in the figure, the better the balance between identity preservation and prompt alignment; the FID scores are labeled on the methods that are compared; ours is closest to the upper right corner with the best FID.

**User Study.** To assess alignment with human perceptual judgment, a user study was conducted, assessing MangaCrafter against a cohort of prominent methods, including SDXL, DiffSensei, and StoryDiffusion. In this study, twenty users made selections to identify the framework that produces the most compelling results, based on a holistic assessment of reference character similarity, prompt consistency, image quality, and storytelling ability. Table 1 tabulates the findings, which reveal a decisive preference for Man-

Table 1: **User study** wherein 20 participants were asked to indicate their best-preferred mangas.

| Method | DiffSensei | StoryDiffusion | SDXL | Ours |
|---|---|---|---|---|
| Percent (%) | 1 | 10 | 21 | **68** |

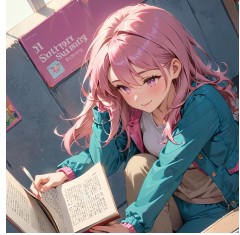 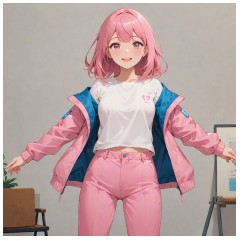 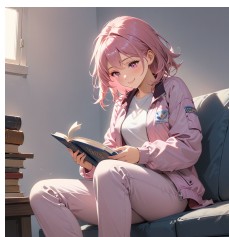

Reference Image      Phase 1 Only      Phase 1 + 2 Only      Phase 1 + 2 + 3

Figure 5: **Qualitative Ablation.** The reference prompt is: "a happy girl, pink eyes, wearing a jacket and trousers". The modification prompt is: "reading a book, eyes down, dutch angle".

gaCrafter, confirming its substantially better quality and ability to generate manga outputs better aligning with human creative intent.

**Ablation Study.** Figure 4, Figure 5 and Table 2 show our ablation study to validate the efficacy of our multi-phase design. The sole inclusion of SRI in Phase 1 yields a strong CLIP-T at the expense of less reference similarity. Adding the PDC in Phase 2 with the exclusion of Phase 3 can drastically improve identity preservation, boosting CLIP-I and dropping DreamSim exceptionally. Crucially, the final results for our complete pipeline (Phase 1+2+3) demonstrate the substantial impact of the "zero-out" strategy, as the CLIP-I and DreamSim scores shift to a much more balanced, meaningfully lower position than for Phase 1+2. This study quantitatively confirms that Phase 3 successfully mitigates the over-similarity problem while retaining excellent identity control, striking an optimal balance for high-quality narrative generation.

## 5 DISCUSSIONS AND LIMITATIONS

Our framework introduces a new paradigm for controllable narrative generation, demonstrating state-of-the-art identity preservation with high prompt fidelity. While our approach shows remarkable robustness, its performance is governed by a set of hyperparameters, including the phase transition timesteps ($T_{phase1}, T_{phase2}$)

Table 2: **Quantitative Ablation.** We evaluated the impact of each phase of MangaCrafter. "Ours (Phase 1 + 2 + 3)" represents the complete pipeline.

| Method | CLIP-T↑ | CLIP-I↑ | DreamSim↓ |
|---|---|---|---|
| SDXL (Baseline) | 0.9045 | 0.8601 | 0.2360 |
| Phase 1 | 0.9135 | 0.8610 | 0.2460 |
| Phase 1 + 2 | 0.8462 | **0.9845** | **0.0058** |
| Phase 1 + 2 + 3 (Ours) | **0.9151** | 0.8983 | 0.1412 |

and the PDC gains ($K_p, K_i, K_d$). However, we have found that our provided default configurations are highly effective across a wide range of scenarios, and the modular nature of the phases allows for intuitive tuning. Additionally, our solution to the multi-character fusion problem involves an explicit multi-pass generation workflow. While this introduces computational overhead, it is a deliberate trade-off that significantly eradicates the identity-bleed issues plaguing concurrent methods. Finally, compared to full-page generation frameworks where a single modification necessitates regenerating the entire page, often with unpredictable results, our work is much more convenient for users as they can quickly change one manga panel just by modifying that panel's prompt while maintaining the original layout, which is much more favoured by users in real-life settings.

## 6 CONCLUSION

We present **MangaCrafter**, a training-free, multi-phased framework for consistent manga generation that addresses the long-standing challenge of preserving character identity while enabling expressive, prompt-driven synthesis. By introducing *Structural Resonance Injection (SRI)* to establish early structural similarity, the *Predictive Drift Controller (PDC)* to dynamically stabilize feature drift, and a strategic zero-out phase to balance identity control with creative flexibility, our method achieves robust identity preservation and high-quality, diverse character synthesis without any fine-tuning. Extensive experiments on the ConsiStory+ benchmark demonstrate that MangaCrafter consistently outperforms state-of-the-art training-free methods in both quantitative metrics and user studies, confirming its superior abilities. Ablations further validate the efficacy of our phased design in striking an optimal balance between similarity, diversity, and visual appeal. Together with a lightweight multi-character preprocessing workflow, MangaCrafter establishes a practical and scalable paradigm for controlled narrative generation in diffusion-based media.

**Story Summary**
A worried girl making a phone call

**Reference Prompts**
[Ceci] a school girl, wearing golden glasses.

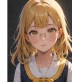

(1) a school girl, wearing golden glasses, looking at phone with a worried expression, chest up, front view, eyes down [Ceci], living room background.
(2) a school girl, wearing golden glasses, holding phone to her ear, anxious look, upper body, eyes up [Ceci], living room background.
(3) a school girl, wearing golden glasses, talking animatedly on the phone, frustrated expression, "I just don't know what to do anymore!", upper body, drone view, eyes looking down [Ceci], living room background.
(4) a school girl, wearing golden glasses, listening carefully to her phone, side view, drone view, wide angle, eyes closed [Ceci], living room background.
(5) a school girl, wearing golden glasses, a small smile of relief while on the phone, "That... that might actually work. Thank you.", low angle, chest up, eyes up right looking above [Ceci], living room background.
(6) a school girl, wearing golden glasses, ending the phone call with calmness, looking relieved and peaceful, exhaling with a relief, eye level, upper body, eyes closed [Ceci], living room background.

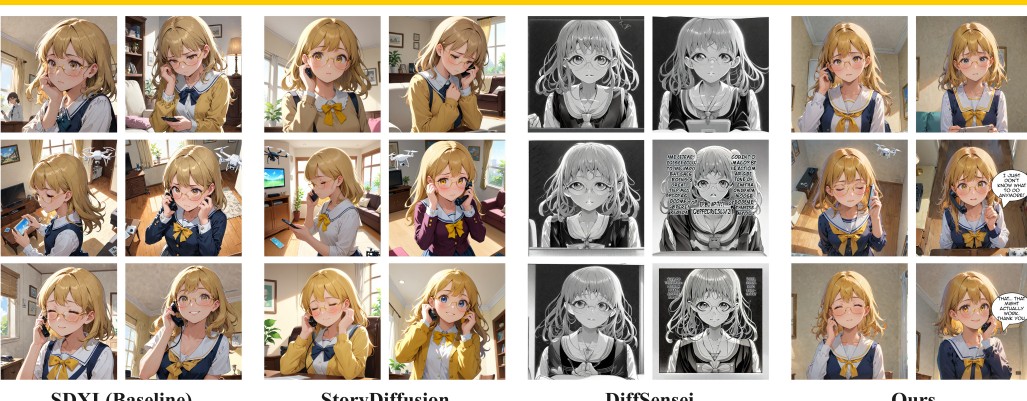

| SDXL(Baseline) | StoryDiffusion | DiffSensei | Ours |

**Story Summary**
A purple astronaut exploring a distant planet with his robot assistant

**Reference Prompts**
[Robo] a white drone.
[Astro] a purple astronaut.

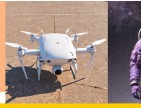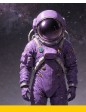

(1) a purple astronaut, standing on a cliff, looking over a vast alien landscape, wide angle, eye level, eyes right. "Let's get it." [Astro], a white drone, by his side, [Robo], distant alien planet background, rocky terrain, strange flora.
(2) a purple astronaut, pointing towards a strange, glowing rock formation, "Robo, get a load of this!", upper body, eyes up right [Astro], distant alien planet background, rocky terrain, strange flora.
(3) a white drone, its optical sensors on its round body glowing as it analyzes the rock, "Scanning... composition unknown, high energy readings detected.", front view [Robo], distant alien planet background, rocky terrain, strange flora.
(4) a purple astronaut, kneeling down, looking at a big rock, his helmet reflecting the light, "Fascinating! This is a major discovery.", low angle, eyes down [Astro], distant alien planet background, rocky terrain, strange flora.
(5) a purple astronaut, walking off towards a mountain range in the distance, "Let's head back!", full body, back view [Astro], a white drone, floating beside him, [Robo], distant alien planet background, rocky terrain, strange flora..

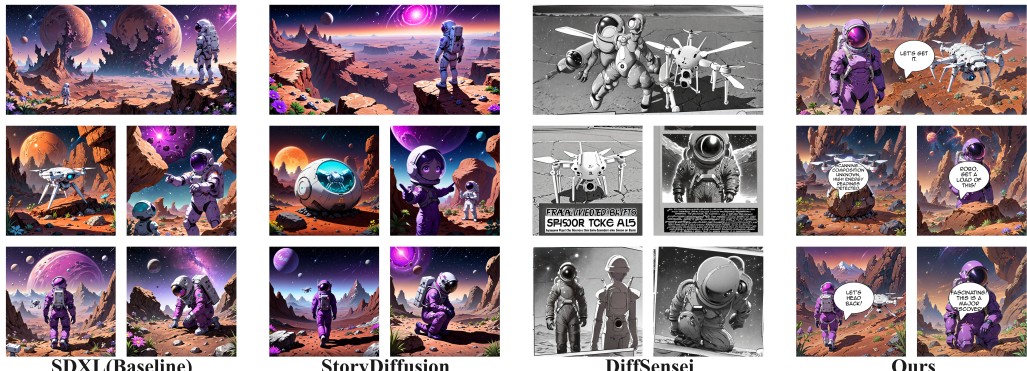

| SDXL(Baseline) | StoryDiffusion | DiffSensei | Ours |

Figure 6: **Additional Qualitative Comparison.** The results presented in this figure compare the outputs of our framework with those from prominent baselines, namely SDXL, StoryDiffusion, and DiffSensei.

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

# APPENDIX

## A  ADDITIONAL QUALITATIVE RESULTS

Further results in Figure 7 and Figure 8 confirms the exceptional performance of our framework in generating complex, multi-character manga pages with a high degree of textual alignment and aesthetic appeal. Our method demonstrates a remarkable capacity to preserve the intrinsic qualities of the base model, a crucial attribute that quantitative metrics often fail to capture. These results affirm our method's state-of-the-art standing in producing coherent, high-quality, and practically useful narrative visuals.

## B  ADDITIONAL QUALITATIVE COMPARISON RESULTS

Qualitative comparisons in Figure 6 and Figure 9 underscore the practical superiority of our method, revealing the significant limitations of current state-of-the-art approaches. While the baseline SDXL (Podell et al., 2023) offers no mechanism for character consistency, even specialized frameworks falter. StoryDiffusion (Zhou et al., 2024a), for example, exhibits poor reference similarity and fails to capture the dynamic storytelling nature of manga. Similarly, DiffSensei (Zhou et al., 2024a) struggles with weak character similarity, producing low-quality, monochrome outputs. In stark contrast, our framework excels, delivering an outstanding balance between robust reference similarity and precise prompt alignment. This synergy results in aesthetically superior images that cohere into compelling narratives, demonstrating a profound leap in storytelling capability. For the purposes of a fair and direct qualitative comparison, ConsiStory (Tewel et al., 2024) and One-Prompt-One-Story (Liu et al., 2025) have been omitted from this evaluation, as their architectures do not natively support image-based conditioning.

## C  IMPLEMENTAION DETAILS

Our training-free framework's multi-phase control process is dynamically configured based on the panel's complexity. For single-character generation, the Structural Resonance Injection (SRI) phase is active for timesteps $t > 850$, with the Predictive Drift Controller (PDC) operating during $800 < t \leq 850$, using gains of $K_p = 0.25$, $K_i = 0.3$, and $K_d = 0.25$. For the more demanding multi-character generation pass, as detailed in Section 3.4, these parameters are intensified for robust identity separation: SRI is active for $t > 675$, the PDC for $575 < t \leq 675$, and controller gains are elevated to $K_p = 0.55$, $K_i = 0.5$, and $K_d = 0.55$.

## D  USER STUDY DETAILS

We conducted a comprehensive user study involving 20 participants. As illustrated in Figure 10, for each task, participants were presented with a complete one-page manga narrative, including the high-level story summary, reference characters, and detailed per-panel prompts. They were then shown several full-page manga generations from our method and competing baselines, and were asked to select the single best entry based on a holistic evaluation of four distinct criteria: Reference Similarity, Textual Consistency, Image Quality, and Storytelling Ability. This robust evaluation protocol was designed to capture the nuanced, multifaceted qualities of a compelling visual narrative that are often missed by quantitative analysis.

## E  USE OF LARGE LANGUAGE MODELS

Large Language Models(LLMs) were only used to polish writing sporadically in the paper.

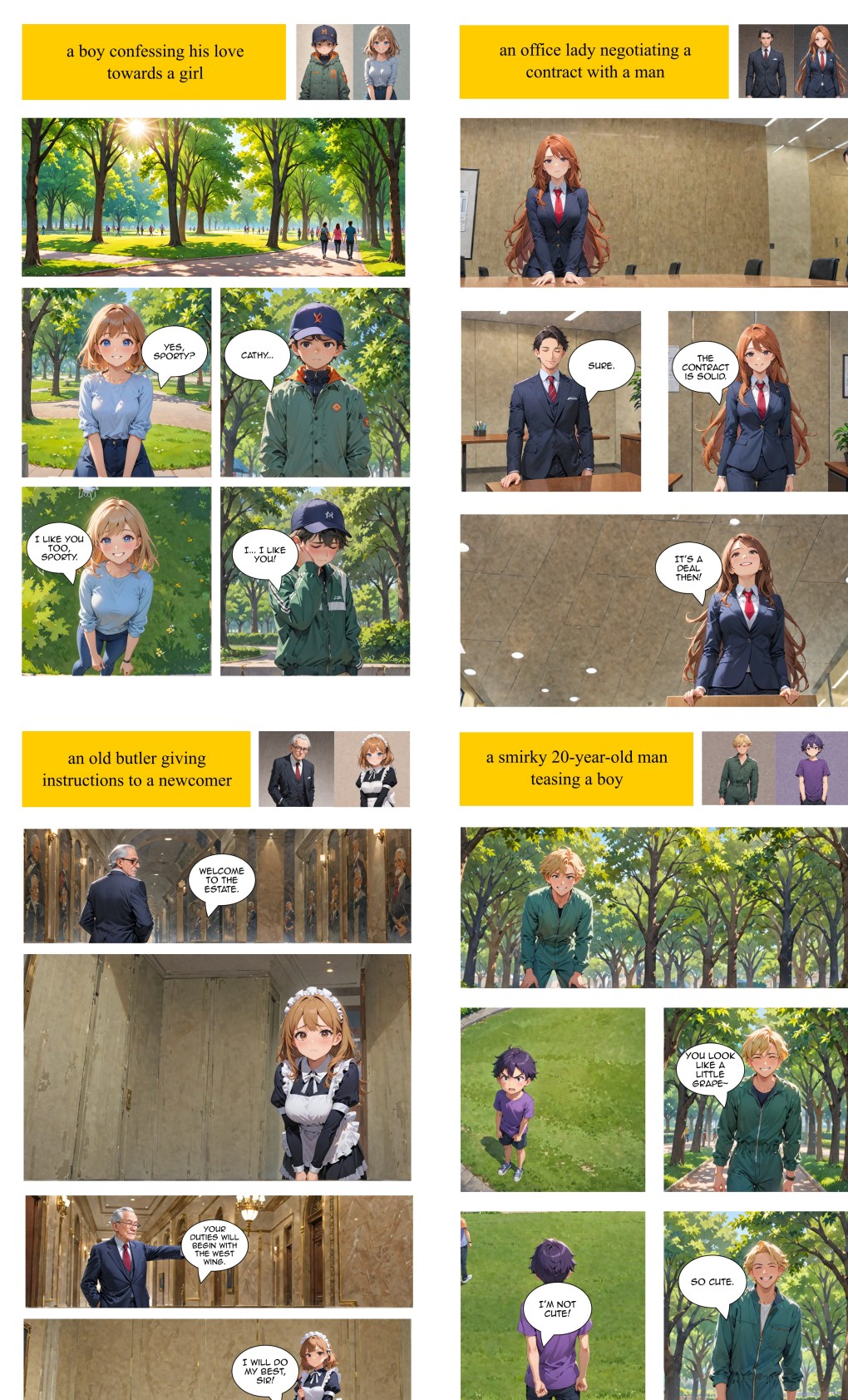

Figure 7: **More Additional Qualitative Results.** Additional results from our method.

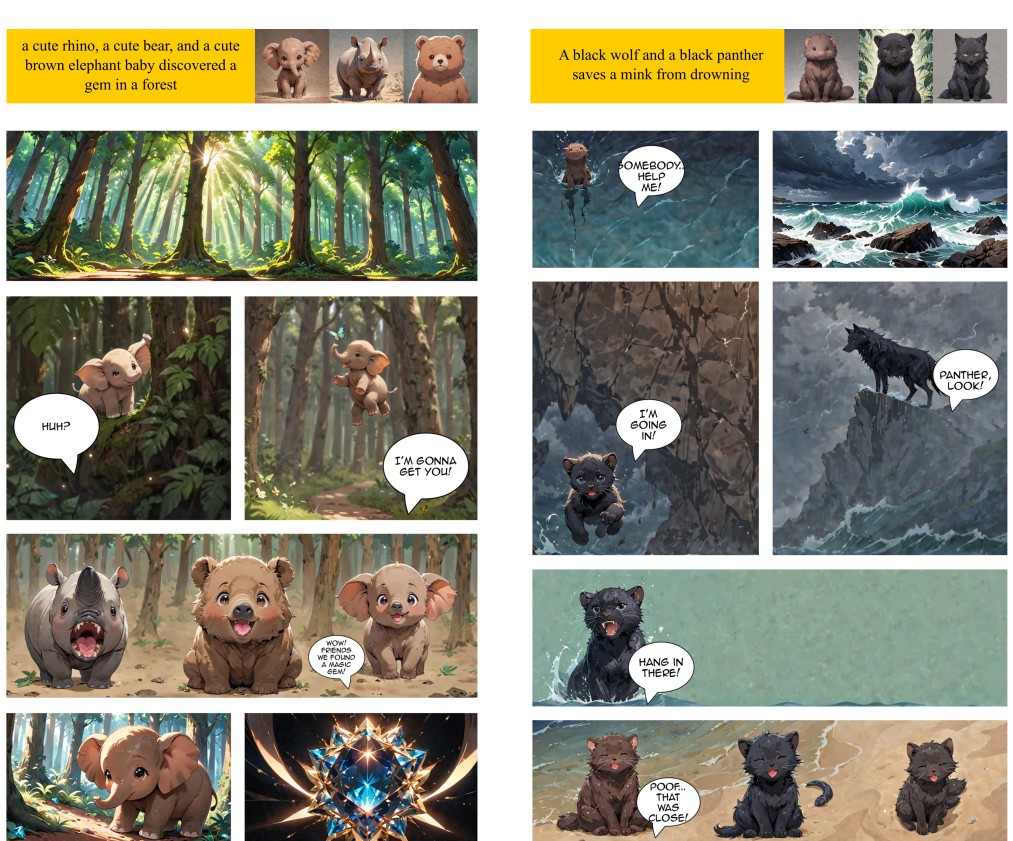

Figure 8: **Additional Qualitative Results On Three Consistent Characters.** Additional results showing our method capable of generating more than two consistent characters within the same manga panel.

**Story Summary**

A girl and a boy meeting at a playground in the morning

**Reference Prompts**

[Billy] a boy, wearing a hoodie.
[Hola] a girl, long hair, wearing a hoodie.

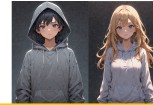

(1) an empty school playground, morning, sun shining, swings, slide, wide angle.
(2) a girl, long hair, wearing a hoodie, walking, waving, upper body [Hola], school playground background.
(3) a girl, long hair, wearing a hoodie, smiling, "Hey! You're here early.", chest up, eyes right looking sideways [Hola], school playground background.
(4) a boy, wearing a hoodie, smiling back, "Couldn't sleep.", chest up, eyes left looking sideways [Billy], school playground background.
(5) a boy, wearing a hoodie, the grey hoodie looks perfect on him, chatting [Billy], a girl, long hair, wearing a hoodie, high angle, eyes down, "Greater weather... Isn't it?" [Hola], school playground background.

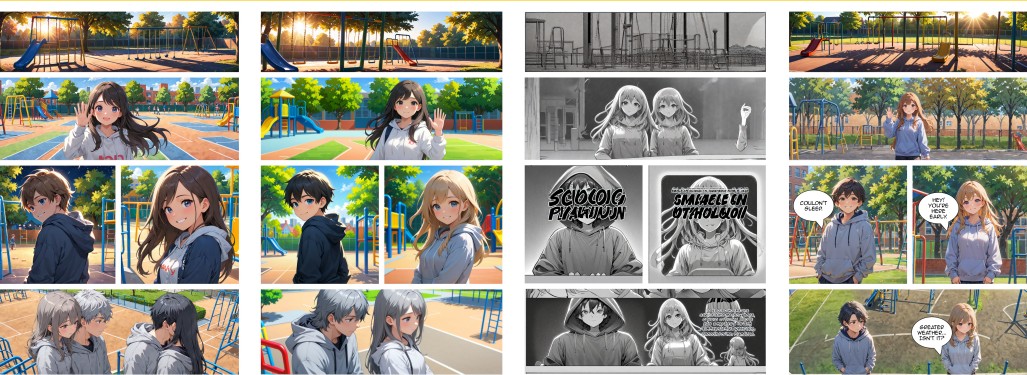

| SDXL(Baseline) | StoryDiffusion | DiffSensei | Ours |
|---|---|---|---|

**Story Summary**

A boy feeding his pet mouse at home

**Reference Prompts**

[Mouse] a cute nerdy mouse.
[Teeny] a teenage boy with black hair.

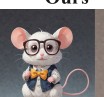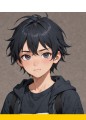

(1) a teenage boy with black hair, holding a small piece of food, looking down affectionately, high angle, eyes down [Teeny], cozy bedroom background.
(2) a cute nerdy mouse, standing on its hind legs, looking up eagerly, low angle, eyes up [Mouse], cozy bedroom background.
(3) a teenage boy with black hair, boy's hand gently offering a small plate of food, "Here you go.", side view, eyes right looking sideways [Teeny], cozy bedroom background.
(4) a cute nerdy mouse, nibbling on the food, eyes down, the little bow tie on the small blue jacket twitches as he eats, high angle [Mouse], cozy bedroom background.
(5) a teenage boy with black hair, smiling, high angle, upper body, eyes down left [Teeny], cozy bedroom background.

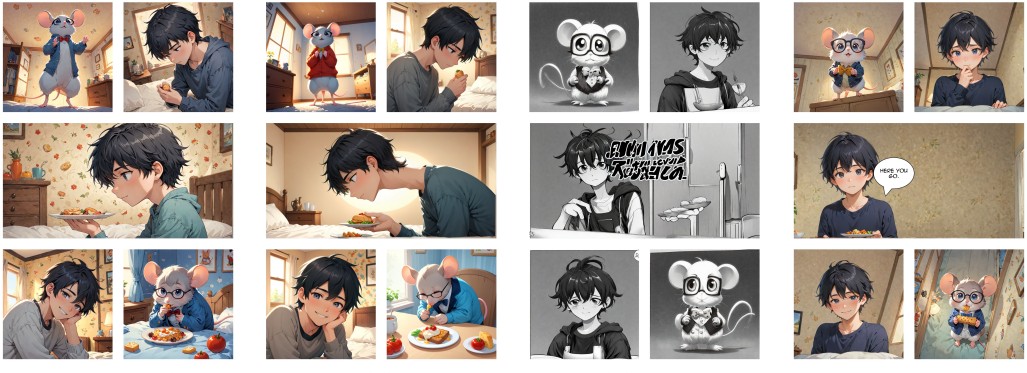

| SDXL(Baseline) | StoryDiffusion | DiffSensei | Ours |
|---|---|---|---|

Figure 9: **More Additional Qualitative Comparison.** Additional results showcasing the difference between our method with SDXL, StoryDiffusion, and DiffSensei.

Figure 10: **User Study Details.** Participants were given clear instructions and criteria for selection.

