# OpenReview forum: "MangaCrafter: Training-Free Consistent Manga Generation via Staged Diffusion"
_ICLR.cc/2026/Conference — ICLR 2026 Conference Withdrawn Submission_

### Official Review · Reviewer_hbcZ · 2025-10-31

**Soundness:** 2
**Presentation:** 2
**Contribution:** 2
**Rating:** 4
**Confidence:** 4

**Summary:**

The authors propose Mangacrafter, a training-free framework for consistent multi character manga generation via diffusion models. Instead of finetuning or textual inversion in the domain of peronalization with diffusion models, they propose a three phase denoising control strategy of SRI, PDC, and Zero out. Their proposed framework achieves high identity consistency and prompt alignment over prior training free methods.

**Strengths:**

- Their proposed three stage framework has an intuitive motivation based on prior works that dive into the relations of how the denoising steps contribute to the final generated output. They anchor the identity early, and introduce prompt alignment in the later stages.
- Applying control theory (PID) to diffusion features is conceptually novel, albeit depending on many hyperparameters.
- The visual results are aesthetically better than its training free baselines.

**Weaknesses:**

- The contributions are mostly engineering refinements over existing attention control/feature blending methods, framed as "phased diffusion". Although applying PID is conceptually novel, there's little theoretical analysis or rigorous justificiation of why PID control is optimal beyond empirical tuning.
- There are no quantitative evaluation on layout or narrative coherence, only identity/prompt metrics. I acknowledge that there might be no such metric that evaluates narrative coherence, but including/proposing something with VLMs may be plausible here. Although the included user study does include a "storytelling ability" criterion, it feels like a missed opportunity to dive deeper into one factor that's crucial in manga.
- The task framing seems out of place. The proposed methods (the 3 phase strategy) seems to heavily modulate images in terms of personalization, not manga generation. The proposed method seems to be able to consistently recreate key elements that ground a subject, but doesn't seem to partake in the consistency of what makes a manga a manga - narrative coherence.
- The authors claim to only use LLMS in polishing the writing sporadically (section E), but this seems to be an understatement. some sections (especially section 3.3 strategic refinement and liberated synthesis) seems to be heavily generated with LLMs. some phrases like "aesthetically superior images" and "profound leap in storytelling capability" borders on marketing and undermines the scientific tone.

**Questions:**

- Is it possible to isolate the 3 phase strategy in the form of the classic personalization task with a wider variety of subject generation? generating a subject concurrently in a manga is less hard in the terms of personalization, since it follows the flow of time (similar backgrounds, same attire, etc). A demonstration of how robust the 3 phase method is in preserving subject identity would be appreciated.

---

### Official Review · Reviewer_uuSu · 2025-10-31

**Soundness:** 2
**Presentation:** 2
**Contribution:** 2
**Rating:** 2
**Confidence:** 5

**Summary:**

This paper introduces MangaCrafter, a training-free framework for generating manga pages with consistent characters across diverse poses, actions, and layouts. The method employs a three-phase process: Structural Resonance Injection (SRI), Predictive Drift Control, and a final refinement phase. Extensive experiments on the challenging ConsiStory+ benchmark demonstrate that this comprehensive framework achieves state-of-the-art results in balancing robust identity preservation with high prompt alignment, surpassing existing training-free methods both quantitatively and in human perception studies.

**Strengths:**

- The introduction of classical control principles into the feature space of diffusion models is somewhat novel.

- The authors conduct various experiments to demonstrate the effectiveness of the proposed method.

**Weaknesses:**

- The paper's presentation is not very clear; although it is sufficient to convey the main idea, it can be distracting at times. For example: (1) When extracting cached reference features, is the Layout HTML file required? Figure 2 suggests that it is, but Section 3.1 does not mention how to input the layout HTML file into the UNet, and SDXL's UNet does not appear to support this input. (2) The process for obtaining the drift d(t) and the blended latent $z_{t}^{blend}$ is unclear.

- The method is highly sensitive to multiple hand-tuned hyperparameters, such as the two phase transition timesteps ($T_{phase1}$ and $T_{phase2}$) and the three complex PID control gains ($K_{p}, K_{i}, K_{d}$), making it difficult to generalize or apply to new models or styles without extensive manual calibration.

- The framework introduces significant computational overhead during inference, particularly due to its multi-pass, multi-character synthesis workflow and the step-wise feature drift calculations of the PID controller, which undermines the claimed efficiency of being training-free.

- Ablation studies in Table 2 reveal that the core control mechanism (SRI + PDC) leads to over-similarity, indicating that the PID controller lacks fine control.

**Questions:**

- The authors are encouraged to include comparisons with strong image-conditioned baselines, such as Flux-kontext, to provide a more comprehensive evaluation of their method.

---

### Official Review · Reviewer_9vns · 2025-10-31

**Soundness:** 2
**Presentation:** 2
**Contribution:** 1
**Rating:** 2
**Confidence:** 4

**Summary:**

This work addresses the problem of consistent character generation in manga creation, which requires maintaining character identity across diverse poses, actions, and panel layouts while preserving narrative flexibility. **MangaCrafter** is a training-free framework that achieves consistent manga generation through a three-phase diffusion process. By front-loading identity anchoring and gradually relaxing constraints, it balances character consistency with narrative flexibility without requiring fine-tuning.

**Strengths:**

1. The three-phase design represents a sophisticated approach to managing the consistency-flexibility trade-off. The transition from aggressive control (SRI) to nuanced guidance (PDC) to complete liberation (Phase 3) is well-motivated.
2. The preprocessing workflow for multi-character panels demonstrates practical utility for real manga creation.

**Weaknesses:**

1. The settings differ for single-character ($t>850$, $800<t≤850$) versus multi-character ($t>675$, $575<t≤675$) generation, but there is no theoretical or empirical justification for these specific values. The performance is likely sensitive to these choices.
2. The method requires generating each character individually at high resolution, background removal, composition, and then a final generation pass. The paper acknowledges this overhead but provides no quantitative analysis of the increased inference time compared to single-character generation or other methods.
3. The baseline is out-of-date, base models such as Flux should be tested.
4. The method is evaluated only on manga generation. Its applicability to other consistent generation tasks (e.g., realistic image sequences, video generation) isn't explored.

**Questions:**

1.  What is the rationale behind the specific phase transition timings and PID gain values? Did you conduct systematic sweeps or is there a principled method for determining these parameters?
2. What guarantees can you provide about the stability of your PID controller in the diffusion feature space?
3. What is the exact computational overhead of the multi-character workflow? How does it scale with the number of characters, and what are the practical limits?
4. Under what conditions does MangaCrafter fail? Are there specific types of character transformations or prompt complexities that challenge the framework?

---

### Official Review · Reviewer_oNwE · 2025-11-01

**Soundness:** 3
**Presentation:** 3
**Contribution:** 3
**Rating:** 8
**Confidence:** 3

**Summary:**

This paper proposes a training-free comic/manga generation framework by dividing timesteps into three phases, serving different purposes suitable to different noise levels. While phase 1 and 3 might not seem new themselves (first get shape, then refine details according to prompt), the dynamic feature blending (PID) at phase 2 sounds interesting and novel.

**Strengths:**

1. The idea to divide timesteps into phases to serve different purposes sounds interesting and technically novel, performing content preservation, style transfer, and text-driven synthesis sequentially.

2. The dynamic feature blending at phase 2 is novel as it continuously adjusts how much style to inject based on measured “drift” from the reference. This helps the transition from injecting reference to injecting prompt more smooth.

3. The training-free method can generalizes better as no tuning is required. And the per-panel generation also provides flexibility for editing. These are practical strengths for easier use.

4. The results look promising and the previous methods for comparison are up-to-date.

**Weaknesses:**

1. As the authors partially point out, there are multiple hyperparameters and design choices, so the heuristic tuning can be expensive.

2. It helps if phase 1+3 results are provided in ablation study, or 1+some very simple blending for smoothness+3, this can help us better understand phase 2 effects.

**Questions:**

Please see Weaknesses. In short, I consider this as a solid work with sound motivation and novelty, thus recommending acceptance. It will be better if concerns in Weaknesses can be addressed, especially for weakness 2.

---

### Note · Authors · 2025-11-13

I have read and agree with the venue's withdrawal policy on behalf of myself and my co-authors.